# Solvent-Free C-3 Coupling of Azaindoles with Cyclic Imines

**DOI:** 10.3390/molecules24193578

**Published:** 2019-10-04

**Authors:** Khadija Belasri, Ferenc Fülöp, István Szatmári

**Affiliations:** 1Institute of Pharmaceutical Chemistry and MTA-SZTE Stereochemistry Research Group, University of Szeged, H-6720 Szeged, Hungary; Khadija.belasri@pharm.u-szeged.hu (K.B.); fulop@pharm.u-szeged.hu (F.F.); 2Institute of Pharmaceutical Chemistry, Interdisciplinary Centre of Excellence, University of Szeged, H-6720 Szeged, Hungary

**Keywords:** cyclic imines, 7-azaindole, 6-azaindole, 4-azaindole, 5-azaindole, *aza*-Friedel-Crafts reaction, microwave reaction

## Abstract

By direct coupling 7-azaindole and cyclic imines, such as 3,4-dihydroisoquinoline, 6,7-dihydrothieno[3,2-*c*]pyridine, 3,4-dihydro-β-carboline, and 4,5-dihydro-*3H*-benz[*c*]azepine, new 3-substituted 7-azaindole derivatives have been synthesized. The reaction was extended to 4-azaindoles and 6-azaindoles, as electron-rich aromatic compounds. The lowest reactivity was observed in the case of C-3 substitution of 5-azaindole. In this case, the aza-Friedel-Crafts reaction took place by using 10 mol % of *p*-toluenesulfonic acid (*p*-TSA) as the catalyst. The role of the acid catalyst can be explained by the different pKa values of the azaindoles. All reactions were performed in solvent-free conditions by using both classical heating and microwave irradiation. In all cases, microwave heating proved to be more convenient to synthesize new C-3-substituted azaindole derivatives.

## 1. Introduction

7-Azaindole is a well-known hinge-binding element in kinase inhibition [1,2,3,4,5]. The N atom of the pyridine ring and the NH group of the pyrrole moiety of 7-azaindole serve as the hydrogen bond acceptor and donor, respectively, to make bidentate hydrogen bonds with the hinge region of the kinase. 7-Azaindole has five modification sites where various substituents can be readily attached to obtain 7-azaindole derivatives with improved activity. Among them, vemurafenib [6], a B-RAF kinase (serine –threonine kinase (STK)) inhibitor, is the first U.S. Food and Drug Administration (FDA)-approved, 7-azaindole-based kinase drug for the treatment of melanoma [7]. Vemurafenib was discovered through lead optimization, starting from a small 7-azaindole fragment, and now it is recognized as the first successful example of a “fragment-based” drug discovery approach [8]. Some derivatives have been developed that target various kinds of kinases, including Janus kinase 3 (JAK3; a cytoplasmic tyrosine kinase (TK)) [9]; colony stimulating factor 1 receptor (CSF1R; a TK receptor) [10]; aurora kinases (STKs) [11]; and Rho-associated, coiled coil-containing protein kinase 1 (ROCK1; STK) [12].

Thanks to the close similarity of azaindoles to the indole skeleton, C-3 functionalization of these compounds has been postulated [13,14]. Most of the methods known already for the synthesis of 3-functionalized 7-azaindole derivatives have applied multistep transformations. Particular efforts have been made to insert another biologically active moiety, such as tetrahydroisoquinoline, into position three. In this case, the synthesis of 1-(7-azaindol-3-yl)-1,2,3,4-tetrahydroisoquinoline involves the coupling of 7-azaindole with N-protected tetrahydroisoquinoline under iron and copper catalysis [15,16]. The catalyst-free direct coupling of partially-saturated cyclic amines and indole as an electron-rich aromatic compound, via a modified *aza*-Friedel-Crafts reaction, has recently been published by our research group [17,18]. The reaction has been extended by using indole-2-carboxylic acid as a substrate, leading to the formation of γ-amino acid derivatives [17].

Our aim was to perform a systematic study of the synthesis of azaindole derivatives (7-, 4-, 5-, and 6-azaindoles), applying the modified aza-Friedel–Crafts reaction using cyclic imine substrates, such as 3,4-dihydroisoquinoline; 6,7-dihydrothieno[3,2-*c*]pyridine; 3,4-dihydro-β-carboline; and 4,5-dihydro-*3H*-benz[*c*]azepine.

## 2. Results

In our initial experiments, 7-azaindole (compound **1**) was reacted with 1.5 equivalent of 3,4-dihydroisoquinoline (compound **2**), which was synthesized using a method from the literature (Scheme 1) [18]. The reaction between compounds **1** and **2** was performed in solvent-free conditions by classical heating (oil bath) at 60 °C (*i*). Based on thin-layer chromatography (TLC), the reaction over a time of 18 h resulted in the formation of a multi-spot reaction mixture; the desired product (compound **3**) was isolated by chromatography, with a yield of 31%. Since the yield was not satisfactory, the reaction was repeated at 80 °C (*ii*). A 10 h reaction led to the formation of compound **3**, with a yield of 49%. Despite increasing the temperature further (100 °C, (*iii*)), the relatively long reaction led to compound **3** only with a poor yield (28%). When the reaction was repeated under microwave irradiation by testing three different reaction conditions (Table 1), 3-(1,2,3,4-tetrahydroisoquinolin-1-yl)-7-azaindole (**3**) was isolated with a yield of 81% after 120 min at 100 °C (Table 1). It is interesting to note that for microwave reactions, 2 equivalents of 3,4-dihydroisoquinoline was applied to provide homogeneity for the reaction mixture.

The reaction was extended by using other cyclic imines, such as 3,4-dihydro-β-carboline (compound **4**) [19]; 6,7-dihydrothieno[3,2-*c*]pyridine (compound **5**) [20]; and 4,5-dihydro-*3H*-benz[*c*]azepine (compound **6**) [21,22]. Reactions were performed by using both oil-bath heating and microwave irradiation (Scheme 1). Table 1 shows that the corresponding products—compounds **7**, **8**, and **9**—were isolated in higher yields when microwave conditions were applied. It can be concluded that 6,7-dihydrothieno[3,2-*c*]pyridine (**5**) was found to be the most reactive cyclic imine. Yields and the applied reaction conditions are summarized in Table 1.

With the optimal reaction conditions in hand, we focused on extending the series of electron-rich aromatic compounds in a modified aza-Friedel-Crafts reaction. Accordingly, 4-azaindole (compound **10**) and 6-azaindole (compound **11**) were also reacted with cyclic imines **2**, **4**, **5**, and **6** (Scheme 2).

Reactions were performed under solvent-free conditions by both classical heating and using microwave irradiation. The new azaindole derivatives, formed in reaction times indicated in Table 2, were isolated and purified by crystallization or column chromatography (see Material and Methods). It can be concluded that the use of microwave irradiation afforded the desired products (compounds **12**–**19**) in higher yields and with significantly shorter reactions, compared with those using oil-bath heating (Table 2).

Since the catalyst-free coupling of 7-, 6-, and 4-azaindoles resulted in the desired C-3 aminoalkylated derivatives, we focused our attention on the aza-Friedel-Crafts reaction of 5-azaindole (compound **20**). In our first experiment, **20** was reacted with 3,4-dihydroisoquinoline (**2**) as representative cyclic imine (Scheme 3). However, even when applying different reaction conditions (classical heating, microwave irradiation) and different temperatures (80 °C (*ii*), 100 °C (*iii*), 120 °C (*iv*)), target compound **21** did not form. There was no conversion at lower temperature (80 °C, 100 °C), while higher temperature (120 °C) resulted in the formation of a multi-spot reaction mixture. Since *p*-toluenesulfonic acid (*p*-TSA) is a frequently applied acid catalyst in the modified three-component Mannich reaction [23,24,25], we decided to examine its effect on the reaction between **20** and 3,4-dihydroisoquinoline. First, 10 mol % of *p*-TSA was tested by using oil-bath conditions. In this case, TLC showed the formation of a new compound, which after isolation (30%) proved to be the desired 5-azaindole derivative **21**. Since the reaction needed a relatively long reaction time (19 h) and resulted in a poor yield, it was repeated by using microwave irradiation. In this case, 100 °C proved to be optimal, and 3-(1,2,3,4-tetrahydroisoquinolin-1-yl)-5-azaindole (**21**) was isolated with a yield of 72% (Table 3).

By applying 10 mol % of *p*-TSA catalyst, C-3 aminoalkylation of 5-azaindole was extended with additional cyclic imines, such as 6,7-dihydrothieno[3,2-*c*]pyridine; 3,4-dihydro-β-carboline; and 4,5-dihydro-*3H*-benz[*c*]azepine (Scheme 3). Reaction conditions and yields are summarized in Table 3. As the data show, there is a difference between the reactivity of 7-, 6-, and 4-azaindoles and that of 5-azaindole—that is, for C-3 aminoalkylation of 5-azaindole an acid catalyst was needed. The lower reactivity of 5-azaindole can be accounted for by its higher pKa value (8.42), compared with those of 6-azaindole (5.61), 4-azaindole (4.85), and 7-azaindole (3.67). The given pKa values were calculated by using the Marvin Sketch software (version 16.12.12.0, calculation module developed by ChemAxon) [26]. The pKa values of azaindoles depend on the resonance stabilization of their protonated forms. The reason why the aza-Friedel-Crafts reaction of 5-azaindole occurs only under acidic conditions is that 5-azaindole has the highest basicity in the series of 4-, 5, 6-, and 7-azaindoles. It is interesting to note that the fastest reactions were achieved with the electron-rich aromatic compound 7-azaindole. This is in complete agreement with its calculated pKa value (3.67), indicating that it is the most acidic compound among the azaindoles studied.

## 3. Materials and Methods

### 3.1. General Methods

Melting points were determined on a Hinotek X-4 (Hinotek Technology Co., Ltd., Ningbo, China) melting point apparatus. Elemental analyses were performed with a Perkin-Elmer 2400 CHNS elemental analyzer (Perkin-Elmer, Waltham, MA, USA) in the Institute of Pharmaceutical Chemistry, University of Szeged. Merck Kieselgel 60F254 plates (Merck Hungary, Budapest, Hungary) were used for the TLC. The microwave reactions were performed with a CEM Discover SP microwave reactor (CEM, Matthwes, NC, USA).

The starting cyclic imines 3,4-dihydroisoquinoline (**2**) [18]; 3,4-dihydro-β-carboline (**4**) [19]; 6,7-dihydrothieno[3,2-*c*]pyridine (**5**) [20]; and 4,5-dihydro-*3H*-benz[*c*]azepine (**6**) [21,22] were synthesized according to the literature.

The ^1^H-and ^13^C-NMR spectra were recorded in CDCl_3_ or [D_6_]DMSO solution in 5 mm tubes at room temperature, on a Bruker Avance II spectrometer (Bruker, Karlsruhe, Germany) at 500 (^1^H) and 125 (^13^C) MHz, with the deuterium signal of the solvent as the lock and TMS as the internal standard. All spectra (^1^H, ^13^C) were acquired and processed with the standard BRUKER software.

#### Procedures

For method A, the cyclic imine 3,4-dihydroisoquinoline (66 mg, 0.50 mmol); 6,7-dihydrothieno[3,2-*c*]pyridine (51 mg, 0.38 mmol); 3,4-dihydro-β-carboline (63 mg, 0.38 mmol) or 4,5-dihydro-*3H*-benz[*c*]azepine (55 mg, 0.38 mmol); and the azaindoles (7-, 6- or 4-azaindole, (30 mg, 0.25 mmol)) were mixed in a 25 mL round bottom flask. The mixture was heated in an oil bath. Reaction conditions are shown in Table 1, Table 2 and Table 3. In the case of 5-azaindole, 10 mol % (4.3 mg, 0.025 mmol) of *p*-TSA as a catalyst was also added.

For method B, the mixture of the cyclic imine 3,4-dihydroisoquinoline (66 mg, 0.50 mmol); 6,7-dihydrothieno[3,2-*c*]pyridine (51 mg, 0.38 mmol); 3,4-dihydro-β-carboline (63 mg, 0.38 mmol) or 4,5-dihydro-*3H*-benz[*c*]azepine (55.3 mg, 0.38 mmol); and the electron-rich aromatic compound (7-, 6-, 4-, or 5-azaindole (30 mg, 0.25 mmol)) were placed in a 10 mL pressurized reaction vial and heated in a microwave reactor, under the conditions given in Table 1, Table 2 and Table 3. In the case of 5-azaindole, 10 mol % of *p*-TSA (4.3 mg, 0.025 mmol) as a catalyst was also added.

### 3.2. Synthesis of New Compounds

#### 3.2.1. Synthesis of 3-(1,2,3,4-Tetrahydroisoquinolin-1-yl)-7-azaindole

Compound **3** (Appendix A) was crystallized from Et_2_O, recrystallized from *i*-Pr_2_O. It was a white solid, with mp: 178–180 °C (Lit.: [16] mp: 172–174°C); yield: 81% (51 mg); ^1^H-NMR ([D_6_] DMSO): δ = 2.73–2.81 (1H, m), 2.91–3.00 (2H, m), 3.10–3.17 (1H, m), 5.29 (1H, s), 6.79 (1H, d, *J* = 7.9 Hz), 6.87 (1H, t, *J* = 7.6 Hz), 6.88–6.93 (1H, m), 6.97 (1H, t, *J* = 5.2 Hz), 7.07–7.16 (2H, m), 7.24 (1H, s), 7.65 (1H, d, *J* = 7.7 Hz), 8.14 (1H, d, *J* = 5.1 Hz), and 11.40 (1H, s); ^13^C-NMR ([D_6_]DMSO): δ = 29.7, 42.3, 54.5, 115.3, 117.5, 118.9, 125.3, 125.8, 126.3, 127.6, 128.6, 129.3, 135.6, 139.2, 142.9, and 149.4. The analysis calculated for C_16_H_15_N_3_ found C: 77.08; H: 6.06; and N: 16.85; found values were C: 77.10; H: 5.98, and N: 16.88.

#### 3.2.2. Synthesis of 3-(1,2,3,4-Tetrahydro-β-carboline-1-yl)-7-azaindole

Product **7** (Appendix A) was crystallized from Et_2_O and recrystallized from *i*-Pr_2_O. It was a white solid, with mp: 219–220 °C; yield: 75% (55 mg); ^1^H-NMR ([D_6_]DMSO): δ = 2.64–2.73 (1H, m), 2.75–2.84 (1H, m), 2.93–3.01 (1H, m), 3.13–3.21 (1H, m), 5.38 (1H, s), 6.87–7.00 (3H, m), 7.15 (1H, d, *J* = 7.8 Hz), 7.30 (1H, s), 7.42 (1H, d, *J* = 7.6 Hz), 7.63 (1H, d, *J* = 8.4 Hz), 8.14 (1H, d, *J* = 4.4 Hz), 10.38 (1H, s,), and 11.50 (1H, s, brs); ^13^C-NMR ([D_6_]DMSO): δ = 22.0, 42.1, 49.7, 107.4, 107.9, 111.6, 118.1, 118.8, 121.3, 123.8, 126.6, 127.2, 135.0, 136.4, 140.3, 140.4, and 143.1. The analysis calculated for C_18_H_16_N_4_ found C: 74.98, H: 5.59, and N: 19.43; the found values were C: 74.91, H: 5.60, and N: 19.45.

#### 3.2.3. Synthesis of 3-(4,5,6,7-Tetrahydrothieno[3,2-*c*]pyridin-4-yl)-7-azaindole

Product **8** (Appendix A) was crystallized from Et_2_O and recrystallized from *i*-Pr_2_O. It was a white solid, with mp: 191–195 °C; yield: 89% (58 mg); ^1^H-NMR ([D_6_]DMSO): δ = 2.74–2.81 (1H, m), 2.85–3.01 (2H, m), 3.12–3.20 (1H, m), 5.25 (1H, s), 6.49 (1H, d, *J* = 6.1 Hz), 6.92–6.98 (1H, m), 7.14 (1H, d, *J* = 5.3 Hz), 7.20–7.24 (1H, m), 7.71 (1H, d, *J* = 9.6 Hz), 8.15 (1H, d, *J* =7.2 Hz), 7.42 (1H, d, *J* = 7.9 Hz), and 11.14 (1H, s,); ^13^C-NMR (CDCl_3_): δ = 26.1, 42.2, 52.3, 115.8, 117.0, 119.0, 121.8, 123.8, 126.4, 128.2, 134.7, 136.6, 143.0, and 149.1 The analysis calculated for C_14_H_13_N_3_S was C: 65.85, H: 5.13, and N: 16.46; found values were C: 65.83, H: 5.15, and N: 16.50.

#### 3.2.4. Synthesis of 3-(2,3,4,5-Tetrahydro-1H-benz[*c*]azepin-1-yl)-7-azaindole

Product **9** (Appendix A)was crystallized from Et_2_O and recrystallized from *i*-Pr_2_O. It was a white solid, with mp: 192–194 °C; yield: 78% (52 mg); ^1^H-NMR ([D_6_]DMSO): δ = 2.87–2.96 (1H, m), 2.97–3.14 (3H, m), 3.30 (2H, s), 5.40 (1H, s), 6.80 (1H, d, *J* = 8.1 Hz), 6.95–7.07 (3H, m), 7.11 (1H, t, *J* = 7.2 Hz), 7.19 (1H, d, *J* = 7.4 Hz), 7.84 (1H, d, *J* = 8.4 Hz), 8.19 (1H, d, *J* = 5.5 Hz), and 11.45 (1H, s); ^13^C-NMR ([D_6_]DMSO): δ = 29.9, 35.5, 49.6, 59.7, 115.2, 115.5, 119.1, 124.5, 126.1, 127.1, 128.0, 128.6, 130.1, 142.5, 143.0, 144.8, 149.8, The analysis calculated for C_17_H_17_N_3_ was C: 77.54, H: 6.51, and N: 15.96; found values were C: 77.60, H: 6.52, and N: 15.98.

#### 3.2.5. Synthesis of 3-(1,2,3,4-Tetrahydroisoquinolin-1-yl)-4-azaindole

Product **12** (Appendix A) was crystallized from Et_2_O and recrystallized from *i*-Pr_2_O. It was a light beige solid, with mp: 166–168 °C; yield: 70% (44 mg); ^1^H-NMR ([D_6_]DMSO): δ = 2.74–2.84 (1H, m), 2.85–2.93 (2H, m), 3.04–3.11 (1H, m), 5.50 (1H, s), 6.94–7.02 (2H, m), 7.06–7.15 (4H, m), 7.74 (1H, d, *J* = 8.1 Hz), 8.32 (1H, d, *J* = 4.6 Hz), and 11.12 (1H, s); ^13^C-NMR ([D_6_]DMSO): δ = 29.7, 41.4, 52.6, 116.8, 119.0, 119.4, 125.6, 126.2, 127.8, 127.9, 129.2, 129.3, 135.6, 139.6, 142.3, and 144.9; The analysis calculated for C_16_H_15_N_3_ was C: 77.08, H: 6.06, and N: 16.85; found values were C: 77.09, H: 5.99, and N: 16.89.

#### 3.2.6. Synthesis of 3-(1,2,3,4-Tetrahydroisoquinolin-1-yl)-6-azaindole

Product **13** (Appendix A) was purified by column chromatography (Dichloro-methane(DCM)/MeOH 9:1) and crystallized from *n*-hexane. It was a light beige solid, with mp: 152–154 °C; yield: 65% (41 mg); ^1^H-NMR ([D_6_]DMSO): δ = 2.72–2.79 (1H, m), 2.89–2.98 (2H, m), 3.08–3.14 (1H, m), 5.29 (1H, s), 6.76 (1H, d, *J* = 8.1 Hz), 6.96 (1H, t, *J* = 5.6 Hz), 7.05–7.15 (2H, m), 7.25 (1H, d, *J* = 7.5 Hz), 7.36 (1H,s), 7.94 (1H, d, *J* = 6.1 Hz), 8.68 (1H, s), and 11.36 (1H, s); ^13^C-NMR ([D_6_]DMSO): δ = 29.8, 42.4, 54.01, 115.0, 118.8, 125.8, 126.3, 127.5, 128.7, 129.3, 130.9, 134.3, 134.8, 135.7, 137.5, and 139.5 The analysis calculated for C_16_H_15_N_3_ was C: 77.08, H: 6.06, and N: 16.85; found values were C: 77.12, H: 5.98, and N: 16.87.

#### 3.2.7. Synthesis of 3-(1,2,3,4-Tetrahydro-β-carboline-1-yl)-4-azaindole

Product **14** (Appendix A) was crystallized from Et_2_O, recrystallized from *i*-Pr_2_O. It was a light brown solid, with mp: 157–160 °C; yield: 69% (50 mg); ^1^H-NMR (500 MHz, ([D_6_]DMSO): δ = 2.86–3.06 (2H, m); 3.37–3.54 (1H, m), 6.04 (1H, s), 6.94–7.09 (2H, m), 7.14–7.29 (2H, m), 7.43–7.58 (2H, m), 7.85 (1H, d, *J* =8.6 Hz), 8.34–8.45 (1H, m), 10.81 (1H, s,), and 11.60 (1H, s, brs). ^13^C-NMR (CDCl_3_): δ = 22.4, 42.1, 49.0, 107.78, 111.4, 117.1, 118.0, 118.9, 119.0, 121.3, 125.6, 127.2, 129.6, 135.4, 135.6, 142.6, 143.0, and 144.0. The analysis calculated for C_18_H_16_N_4_ was C: 74.98, H: 5.59, and N: 19.43; found values were C: 74.97, H: 5.62, N: 19.43.

#### 3.2.8. Synthesis of 3-(1,2,3,4-Tetrahydro-β-carboline-1-yl)-6-azaindole

Product **15** (Appendix A) was purified by column chromatography (DCM/MeOH 3:1) and crystallized from *n*-hexane. It was a light brown solid, with mp: 217–219 °C; yield: 62% (45 mg); ^1^H-NMR ([D_6_]DMSO): δ = 2.65–2.84 (2H, m), 2.93–3.02 (1H, m), 3.11–3.19 (1H, m), 6.92–6.99 (2H, m), 5.43 (1H, s), 7.18 (1H, d, *J* =7.6 Hz), 7.24(1H, d, *J* = 5.4 Hz), 7.38–4.44 (1H, m), 7.94 (1H, d, *J* = 5.6 Hz), 8.70 (1H, s), 10.37 (1H, s,), and 11.44 (1H, s, brs); ^13^C-NMR ([D_6_]DMSO): δ = 22.6, 42.4, 49.8, 108.4, 111.52, 114.62, 118.02, 118.60, 120.8, 127.3, 128.9, 131.1, 134.3, 134.9, 136.0, 136.2, and 137.7. The analysis calculated for C_18_H_16_N_4_ was C: 74.98, H: 5.59, and N: 19.43; found values were C: 74.96, H: 5.61, and N: 19.44.

#### 3.2.9. Synthesis of 3-(4,5,6,7-Tetrahydrothieno[3,2-*c*]pyridin-4-yl)-4-azaindole

Product **16** (Appendix A) was crystallized from Et_2_O and recrystallized from *i*-Pr_2_O. It was a white solid, with mp: 179–182 °C; yield: 85% (55 mg); ^1^H-NMR ([D_6_]DMSO): δ = 2.73–2.82 (2H, m), 2.90–2.98 (1H, m), 3.11–3.15 (1H, m), 5.25 (1H, s), 6.50 (1H, d, *J* = 4.9 Hz), 6.91 (1H, t, *J* = 7.4 Hz), 7.05 (1H, t, *J* = 7.6 Hz), 7.07 (1H, s), 7.13 (1H, d, *J* = 5.1 Hz), 7.34 (1H, t, *J* = 7.9 Hz), 7.42 (1H, d, *J* = 7.9 Hz), 10.85 (1H, s), 8.19 (1H, s, brs), 11.52 (1H, s,), and 16.78 (1H, s, brs); ^13^C-NMR (CDCl_3_): δ = 25.6, 41.3, 50.6, 117.1, 118.0, 118.5, 121.9, 126.3, 126.7, 129.3, 134.7, 135.8, 143.0, and 144.4. The analysis calculated for C_14_H_13_N_3_S was C: 65.85, H: 5.13, N: 16.46; found values were C: 65.82, H: 5.12, and N: 16.48.

#### 3.2.10. Synthesis of 3-(4,5,6,7-Tetrahydrothieno[3,2-*c*]pyridin-4-yl)-6-azaindole

Product **17** (Appendix A) was purified by column chromatography (DCM/MeOH 9:1) and crystallized from *n*-hexane. It was a white solid, with mp: 167–169 °C; yield: 75% (49 mg); ^1^H-NMR ([D_6_]DMSO): δ = 2.71–2.80 (1H, m), 2.82–2.99 (2H, m), 3.09–3.16 (1H, m), 5.23 (1H, s), 6.46 (1H, d, *J* = 5.1 Hz), 7.13 (1H, d, *J* =5.6 Hz), 7.25–7.40 (2H, m), 7.98 (1H, d, *J* = 5.4 Hz), 8.68 (1H, s), and 11.35 (1H, s); ^13^C-NMR (CDCl_3_): δ = 26.0, 42.2, 51.8, 114.2, 118.3, 122.0, 126.1, 127.4, 131.3, 133.8, 134.1, 134.8, 136.4, and 138.2. The analysis calculated for C_14_H_13_N_3_S was C: 65.85, H: 5.13, and N: 16.46; found values were C: 65.83, H: 5.15, and N: 16.45.

#### 3.2.11. Synthesis of 3-(2,3,4,5-Tetrahydro-*1H*-benzo[*c*]azepin-1-yl)-4-azaindole

Product **18** (Appendix A) was crystallized from Et_2_O and recrystallized from *i*-Pr_2_O. It was a light beige solid, with mp: 162–165 °C; yield: 68% (45 mg); ^1^H-NMR (500 MHz, ([D_6_]DMSO): δ = 2.95–3.03 (1H, m), 3.18–3.29 (3H, m), 3.30–3.40 (2H, m), 6.06 (1H, s), 6.85 (1H, d, *J* = 7.4 Hz), 7.06 (1H, t, *J* = 7.9 Hz), 7.18–7.25 (2H, m), 7.29 (1H, d, *J* = 7.4 Hz), 7.60 (1H, s), 7.90 (1H, d, *J* = 8.4 Hz), 8.35 (1H, d, *J* = 4.6 Hz), and 11.76 (1H, s); ^13^C-NMR (100 MHz, [D6]DMSO): δ = 25.9, 34.0, 49.4, 57.5, 111.4, 117.6, 119.8, 126.6, 128.7, 128.8, 129.3, 129.4, 130.0, 138.0, 142.5, 142.9, and 144.1. The analysis calculated for C_17_H_17_N_3_ was C: 77.54, H: 6.51, and N: 15.96; found values were C: 77.53, H: 6.53, and N: 15.99.

#### 3.2.12. Synthesis of 3-(2,3,4,5-Tetrahydro-*1H*-benzo[*c*]azepin-1-yl)-6-azaindole

Product **19** (Appendix A) was purified by column chromatography (DCM/MeOH 9:1) and crystallized from *n*-hexane. It was a light beige solid, with mp: 154–156 °C; yield: 64% (42 mg); ^1^H-NMR (500 MHz, ([D_6_]DMSO): δ = 2.91–2.29 (1H, m), 3.13–3.30 (2H, m), 5.89 (1H, s), 6.86 (1H, d, *J* = 7.9 Hz), 7.08 (1H, t, *J* = 7.4 Hz), 7.17–7.30 (3H, m), 7.48 (1H, d, *J* = 5.3 Hz), 7.53 (1H,s), 8.07 (1H, d, *J* = 5.3 Hz), 8.81 (1H, s), and 11.86 (1H, s); ^13^C-NMR (100 MHz, [D_6_]DMSO): δ = 33.8, 34.3, 49.0, 50.1, 50.8, 58.2, 114.6, 126.7, 126.9, 128.6, 128.9, 129.4, 129.7, 130.3, 130.5, 131.0, 134.2, 135.4, 138.2, and 142.6. The analysis calculated for C_17_H_17_N_3_ was C: 77.54, H: 6.51, and N: 15.96; found values were C: 77.56, H: 6.50, and N: 15.94.

#### 3.2.13. Synthesis of 3-(1,2,3,4-Tetrahydroisoquinolin-1-yl)-5-azaindole

Product **20** (Appendix A) was purified by column chromatography (DCM/MeOH 9:1) and crystallized from *n*-hexane. It was a light beige solid, with mp: 118–119 °C; yield: 72% (46 mg); ^1^H-NMR (500 MHz, ([D_6_]DMSO): δ = 2.88–3.39 (4H, m), 6.35 (1H, s), 6.92 (1H, d, *J* = 7.8 Hz), 7.27–7.49(3H, m), 7.80–7.92 (1H, m), 8.03 (1H, d, *J* = 7.1 Hz), 8.45 (1H, d, *J* = 7.1 Hz), 9.14(1H, s), and 13.25 (1H, s); ^13^C-NMR (100 MHz, [D_6_]DMSO): δ = 29.7, 42.3, 54.1, 107.3, 118.8, 123.5, 125.8, 125.9, 126.4, 127.6, 129.3, 135.6, 139.3, 140.3, 140.5, and 143.6. The analysis calculated for C_16_H_15_N_3_ was C: 77.08, H: 6.06, and N: 16.85; found values were C: 77.02, H: 5.98, and N: 16.88.

#### 3.2.14. Synthesis of 3-(1,2,3,4-Tetrahydro-β-carboline-1-yl)-5-azaindole

Product **22** (Appendix A) was purified by column chromatography (DCM/MeOH 3:1) and crystallized from *n*-hexane. It was a light brown solid, with mp: 187–189 °C; yield: 69% (50 mg); ^1^H-NMR (500 MHz, ([D_6_]DMSO): δ = 2.75–2.82 (1H, m), 2.84–2.91 (1H, m), 3.07–3.13 (1H, m), 3.22–3.27 (1H, m), 5.63 (1H, s), 6.97–7.03 (2H, m), 7.18 (1H, d, *J* = 6.6 Hz), 7.32–7.38 (2H, m), 7.47 (1H, d, *J* = 5.4 Hz), 8.11 (1H, d, *J* = 4.5 Hz), 8.49 (1H, s,), 10.53 (1H, s,), and 11.50 (1H, s); ^13^C-NMR (100 MHz, [D_6_]DMSO): δ = 22.0, 42.1, 49.6, 107.3, 107.9, 111.6, 115.2, 118.1, 118.8, 121.3, 123.8, 126.6, 127.2, 135.0, 136.4, 140.3, 140.4, and 143.1. The analysis calculated for C_18_H_16_N_4_ was C: 74.98, H: 5.59, and N: 19.43; found values were C: 74.93, H: 5.58, and N: 19.46.

#### 3.2.15. Synthesis of 3-(4,5,6,7-Tetrahydrothieno[3,2-*c*]pyridin-4-yl)-5-azaindole

Product **23** (Appendix A) was purified by column chromatography (DCM/MeOH 9:1) and crystallized from *n*-hexane. It was a light beige solid, with mp: 113–115 °C; yield: 80% (52 mg); ^1^H-NMR ([D_6_]DMSO): δ = 2.80–3.23 (4H, m), 5.26 (1H, s), 6.48 (1H, d, *J* = 5.1 Hz), 6.98–7.40 (3H, m), 8.08 (1H, d, *J* = 5.5 Hz), 8.16 (1H, s), and 11.24 (1H, s); ^13^C-NMR ([D_6_]DMSO): δ = 25.8, 42.4, 52.0, 107.3, 117.7, 122.4, 123.6, 125.3, 126.7, 134.4, 137.9, 140.3, 140.4, and 143.3. The analysis calculated for C_14_H_13_N_3_S was C: 65.85, H: 5.13, and N: 16.46; found values were C: 65.84, H: 5.14, and N: 16.49.

#### 3.2.16. Synthesis of 3-(2,3,4,5-Tetrahydro-*1H*-benzo[*c*]azepin-1-yl)-5-azaindole

Product **24** (Appendix A) was purified by column chromatography (DCM/MeOH 9:1) and crystallized from *n*-hexane. It was a light beige solid, with mp: 142–145 °C; yield: 65% (43 mg); ^1^H-NMR ([D_6_]DMSO): δ = 2.82–3.09 (6H, m), 5.40 (1H, s), 6.81 (1H, d, *J* = 6.7 Hz), 6.93–7.02 (1H, m), 7.07–7.14 (3H, m), 7.19 (1H, d, *J* = 7.3 Hz), 7.35 (1H, d, *J* = 7.3 Hz), 8.12 (1H, d, *J* = 5.7 Hz), 8.72 (1H, s), 11.27 (1H, brs); ^13^C-NMR (CDCl_3_): δ = 30.2, 35.8, 53.4, 106.8, 117.8, 123.6, 124.3, 126.1, 127.2, 127.7, 129.3, 130.0, 140.3, 140.7, 142.2, 143.2, and 144.0. The analysis calculated for C_17_H_17_N_3_ was C: 77.54, H: 6.51, and N: 15.96; found values were C: 77.57, H: 6.52, and N: 15.94.

## 4. Conclusions

To conclude, the modified aza-Friedel–Crafts reaction was applied for the C-3 substitution of azaindoles. The reactions were achieved by using 3,4-dihydroisoquinoline; 6,7-dihydrothieno[3,2-*c*]pyridine; 3,4-dihydro-β-carboline; and 4,5-dihydro-*3H*-benz[*c*]azepine as imine substrates. The reactions of 4-azaindole and 6-azaindole with similar reactivity led to the formation of new 3-isoquinolyl-, 3-thieno[3,2-*c*]pyridyl-, 3-β-carbolinyl-, and 3-benz[*c*]azepinyl-azaindole derivatives. Starting from 5-azaindole, the modified aza-Friedel–Crafts reaction could only be performed by using 10 mol % of *p*-TSA as a catalyst. Systematic correlation was found between the reactivity of azaindoles and their pKa values. This latter observation allows us to explain why the direct coupling of 5-azaindole works only under acidic conditions. Namely, 5-azaindole (pKa = 8.42) has the highest basicity in the series of 4-, 5-, 6-, and 7-azaindoles. It is important to note that all reactions could be accelerated by using microwave irradiation.

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
