# Peer review of "Solvent-Free C-3 Coupling of Azaindoles with Cyclic Imines"

_molecules, 2019, doi:10.3390/molecules24193578_

Round 1
Reviewer 1 Report
The authors describe a simple and sustainable route to prepare several azaindole compounds. This scaffold is highly interesting for medicinal chemistry and thus the work presented constitutes an important tool to functionalize these compounds at C-3.
The work is well planned however some improvements are needed before acceptance:
in table 1, a and b are not identified - I suppose it is microwave and oil heating conditions.. there is a typo in "temperature" in all document including tables - must be corrected lines 84 to 97 - the bold is missing in the compounds identification in scheme 3 there is no correspondence between the ii, iii, iv and the text , I did not understand the meaning..please refer in the correspondent text for a facile understanding Line 108 - pKa values were calculated based on the yield. I do not agree with this statement. Additionally there are documented in the literature pKa values calculated for azaindoles that can be mentioned as a reference. all nmr data should have the "J" in italic 272 - I did not receive any supplementary information These references should be added: Org. Lett. 2016, 18, 3250−3253; Molecules 2018, 23, 2673; doi:10.3390/molecules23102673; Mérour, J.Y.; Buron, F.; Plé, K.; Bonnet, P.; Routier, S. The azaindole framework in the design of kinaseinhibitors. Molecules 2014, 19, 19935–19979. After these modifications I recommend that the manuscript is accepted.
Author Response
Reflections on Referee #1’s comments:
“In table 1, a and b are not identified - I suppose it is microwave and oil heating conditions. there is a typo in "temperature" in all document including tables - must be corrected lines 84 to 97 - the bold is missing in the compounds identification in scheme 3 there is no correspondence between the ii, iii, iv and the text , I did not understand the meaning.please refer in the correspondent text for a facile understanding”
Footnotes a and b have been inserted, the typo (concerning the temperature) and the missing bold identifications for the compounds have been checked and corrected. ii), iii) and iv) have been inserted in the text.
“pKa values were calculated based on the yield.”
The sentence has been corrected to:
“The given pKa values were calculated by using the Marvin–Sketch software (version 16.12.12.0, calculation module developed by ChemAxon)” and is highlighted with yellow background.
“All nmr data should have the "J" in italic”
Letters J concerning the coupling constants have been corrected to italic.
“These references should be added: Org. Lett. 2016, 18, 3250−3253; Molecules 2018, 23, 2673; doi:10.3390/molecules23102673; Mérour, J.Y.; Buron, F.; Plé, K.; Bonnet, P.; Routier, S. The azaindole framework in the design of kinase
Molecules 2014, 19, 19935–19979.”
The suggested citations have been inserted.
Reviewer 2 Report
The manuscript by Belasri et al. describes a simple, but effective synthetic approach to a very interesting type of bis-heterocyclic scaffolds that represent versatile building blocks for drug-like compounds. The described C-3 functionalization of various azaindoles with different cyclic imines could be achieved under "green" conditions, i.e. with solvent-free mixtures (and in most cases in the absence of any catalyst). In the case of 5-azaindole, addition of a catalytic amount of acid was necessary which could be explained by the higher pKa value of this synthon. Moreover, it was nicely demonstrated how the synthetic utility of these transformation benefits from the replacement of conventional heating with microwave irradiation.
The paper is carefully written and well referenced. All new compounds are adequately characterized. Of course, it would have been nice to have NMR signal assignments included, at least for the 1H NMR signals (which should be feasible with a 500 MHz instrument).
There are some minor issues to be addressed:
*) line 51: "method (Scheme 1).15" should read "method (Scheme 1) [15]."
*) line 69, Table 1:
1) a footnote (such as that in Table 2) is missing, explaining the meaning of "a" (conventional heating) and "b" (microwave irradiation)
2) both of the "Temperature" columns are labelled as "a", but there should be oine "a" and one "b"
*) lines 84-97: all compound numbers should be printed in bold
*) line 100, Table 3:
1) There is a footnote for "a", "b", and "c", but there is no table entry for "c"
2) Why is the "Temperature" column always labelled as "a"?
*) line 108: what does "calculated by yields" mean?
*) line 136/130: "imine 3,4-dihydroisoquinoline (66,61 mg,"
1) please use uniform precision for amounts (compare methods A/B)
2) please use a decimal point instead of a comma
*) line 144: please add (in parentheses) the mp that was reported in refs [12,13]: 172-174 °C
*) line 146 (and following entries): please report coupling constants with a precision of one decimal digit
*) line 269: "underwent only under acidic conditions." should be better "works only under acidic conditions."
*) line 329 (ref. 16): the correctly abbreviated name of the journal is "Bioorg. Med. Chem." and should be printed in italic. The word "Bioorg." should be deleted from line 327
Author Response
Reflections on Referee #2’s comments:
“method (Scheme 1).15" should read "method (Scheme 1) [15].”
The citation in line 51 has been corrected.
“ a footnote (such as that in Table 2) is missing, explaining the meaning of "a" (conventional heating) and "b" (microwave irradiation) and both of the "Temperatue" columns are labelled as "a", but there should be one "a" and one "b" ”
The footnotes have been inserted and the Temperature are also labelled according to “a” and "b".
“Lines 84-97: all compound numbers should be printed in bold”
The numbering of compounds have been changed to bold in the text in question.
“Line 100, Table 3: There is a footnote for "a", "b", and "c", but there is no table entry for "c" and Why is the "Temperature" column always labelled as "a"?”
The footnote “c” has been inserted!, the typo in temperature has been corrected.
“Line 108: what does "calculated by yields" mean?”
The sentence has been corrected to:
“The given pKa values were calculated by using the Marvin–Sketch software (version 16.12.12.0, calculation module developed by ChemAxon)” and is highlighted with yellow background.
“Line 136/130: "imine 3,4-dihydroisoquinoline (66,61 mg," please use uniform precision for amounts (compare methods A/B) and please use a decimal point instead of a comma”
The precision of amount has been unified and the amount of 3,4-dihydroisoquinoline has been corrected.
Lline 144: please add (in parentheses) the mp that was reported in refs [12,13]: 172-174 °C”
The literature mp for compound 3 has been inserted.
“Line 146 (and following entries): please report coupling constants with a precision of one decimal digit”
The coupling constants have been corrected to a precision of one decimal digit.
“Line 269: "underwent only under acidic conditions." should be better "works only under acidic conditions." ”
The sentence was corrected accordingly.
“Line 329 (ref. 16): the correctly abbreviated name of the journal is " Med. Chem." and should be printed in italic. The word "Bioorg." should be deleted from line 327”
The abbreviation has been corrected and the word "Bioorg." has been shifted into the correct place.
Reviewer 3 Report
Comments for the authors:
The paper is an extension of the authors’ earlier work (Tetrahedron Lett., 2013, reference 14), but this time applied to 3-substituted azaindoles. Azaindoles have not been reported previously by the method given in this paper. However, compound 3 is not new, although it is specified in the Experimental Section as being new. See compound 5r in reference 13 by M. Ghobrial et al. J. Org. Chem. 2011, 76, 8781-8793. Credit should be ascribed to those authors. The other compounds appear to be new as stated by the authors of this paper.
In connection with the acid-catalyzed procedure for the 5-azaindoles, mention should be made of the work of Xie and coworkers, Tetrahedron Lett. 2018, 59, 457-461. Xie et al made the corresponding indoles using a lower temperature aqueous method and methanesulfonic acid catalysis. Going beyond that, it may be possible to make the azaindoles reported in this paper by the milder method of Xie and coworkers or by using their method under microwave conditions in a relatively short time. This shouldn’t take much time to check out. This is likely more relevant for the 5-azaindole experiments.
Rationale for the preparation of 7-azaindoles is given in terms of the fragment-based synthesis of a marketed drug containing a 7-azaindole component. The paper outlines syntheses of other azaindoles. Perhaps there is a drug-related or other rationale that can be presented in connection with reporting the preparation of 4, 5, and 6-azaindole derivatives. (See, for example, Chatterji, M. Antimicrobial Agents and Chemotherapy, 2014, 58, 5325-5331).
Has biological testing been done on any of the products reported in the paper? Since a biological justification for the reported syntheses has been given, it would be relevant to state whether biological testing has been done and if any of such results hold promise.
The notations in the tables should be specified under each table for I, ii and iii (or are these notations even needed?). Table 2, product 18 has a 100 oC with an ii, but it appears it should be iii. Where is footnote c indicated within Table 3? it appears that the last 4 entries should each be labeled as c.
The discussion of relative acidities and basicities could be clarified by adding a table of pKa values of the various compounds and the effects on reactivity. References should be given for sources of pKa values. The statement “The pKa values were calculated by yields” needs explanation. An expansion of the statement concerning the use of Marvin Sketch would be helpful concerning the relative acidities of the azaindoles. Have any of the experiments for other azaindoles been tried using the acid catalyst? If so, is there an effect on the product yields? The assumption is that there would be no effect. Have acid catalysts other than p-TSA been tried? Is the p-TSA used because it gives the highest yields compared to other catalysts?
The microwave method should have details listed concerning power levels used and internal pressures.
The authors comment on the higher yields achieved in the microwave experiments. Are the higher yields related to higher internal pressures in the microwave experiments?
The name of the analyst or analytical laboratory for elemental analyses should be listed in the Experimental Section.
Omit the word “only” in the abstract. Correct the spelling of temperature in the tables, also Marvin Sketch.
Author Response
Reflections on Referee #3’s comments:
“The paper is an extension of the authors’ earlier work (Tetrahedron Lett., 2013, reference 14), but this time applied to 3-substituted azaindoles. Azaindoles have not been reported previously by the method given in this paper. However, compound 3 is not new, although it is specified in the Experimental Section as being new. See compound 5r in reference 13 by M. Ghobrial et al. J. Org. Chem. 2011, 76, 8781-8793. Credit should be ascribed to those authors. The other compounds appear to be new as stated by the authors of this paper.”
By inserting the corresponding citation Ref. 16 with the literature melting point in the experimental part the state that compound 3 has already been described is highlighted.
“In connection with the acid-catalyzed procedure for the 5-azaindoles, mention should be made of the work of Xie and coworkers, Tetrahedron Lett. 2018, 59, 457-461. Xie et al made the corresponding indoles using a lower temperature aqueous method and methanesulfonic acid catalysis. Going beyond that, it may be possible to make the azaindoles reported in this paper by the milder method of Xie and coworkers or by using their method under microwave conditions in a relatively short time. This shouldn’t take much time to check out. This is likely more relevant for the 5-azaindole experiments.”
There were two main reasons why p-TSA was chosen as acid catalyst. First, it is an unexpensive, often used acid catalyst in the Mannich reaction. Second, in one of our previous work, when 2-, or 1-naphthol were aminoalkylated with ammonia in the presence of glyoxylic acid, p-TSA were needed to isolate the desired substituted glycine derivatives (Csütörtöki, R.; Szatmári, I.; Mándi, A.; Kurtán, T.; Fülöp, F. Synlett, 2011, 1940.).
“Rationale for the preparation of 7-azaindoles is given in terms of the fragment-based synthesis of a marketed drug containing a 7-azaindole component. The paper outlines syntheses of other azaindoles. Perhaps there is a drug-related or other rationale that can be presented in connection with reporting the preparation of 4, 5, and 6-azaindole derivatives. (See, for example, Chatterji, M. Antimicrobial Agents and Chemotherapy, 2014, 58, 5325-5331).”
“Has biological testing been done on any of the products reported in the paper? Since a biological justification for the reported syntheses has been given, it would be relevant to state whether biological testing has been done and if any of such results hold promise.”
In frame of another project the biological testing of our products are planned. We wish to present those results in another publication.
“The notations in the tables should be specified under each table for I, ii and iii (or are these notations even needed?). Table 2, product 18 has a 100 oC with an ii, but it appears it should be iii. Where is footnote c indicated within Table 3? it appears that the last 4 entries should each be labeled as c. ”
The footnote “c” has been inserted and the typo in temperature has been corrected.
“The discussion of relative acidities and basicities could be clarified by adding a table of pKa values of the various compounds and the effects on reactivity. References should be given for sources of pKa values. The statement “The pKa values were calculated by yields” needs explanation. An expansion of the statement concerning the use of Marvin Sketch would be helpful concerning the relative acidities of the azaindoles. Have any of the experiments for other azaindoles been tried using the acid catalyst? If so, is there an effect on the product yields? The assumption is that there would be no effect. Have acid catalysts other than p-TSA been tried? Is the p-TSA used because it gives the highest yields compared to other catalysts?”
The sentence has been corrected to:
“The given pKa values were calculated by using the Marvin–Sketch software (version 16.12.12.0, calculation module developed by ChemAxon)” and is highlighted with yellow background.
According to our results, for 7-, 6- and 4-azaindols the yields were satisfactory, so we felt the using of acid catalyst is not indicated. While in case of 5-azaindole applying p-TSA led to acceptable yields, other catalyst has not been tried.
“The microwave method should have details listed concerning power levels used and internal pressures.”
In case of the microwave reactions the only one parameter that can be set is the temperature. If the system reach it the other parameters (power, pressure) change every moments to keep the desired temperature.
“The authors comment on the higher yields achieved in the microwave experiments. Are the higher yields related to higher internal pressures in the microwave experiments? ”
The higher yields achieved by using microwave conditions cannot be explained by the higher internal pressure. This latter parameter changing (increasing) dramatically only in case of using solvents. In solvent-free conditions the changing of the internal pressure can be neglected. The positive effect of the microwave irradiation on the yields can mainly be originated from the special “point heating” in the microwave reactor (Kappe, C. O.; Pieber, B.; Dallinger, D. Angew. Chem. Int. Ed., 2013, 1088.)
“The name of the analyst or analytical laboratory for elemental analyses should be listed in the Experimental Section. ”
The laboratory for elemental analyses is inserted! Detailed information about the used instruments are also included.
“Omit the word “only” in the abstract. Correct the spelling of temperature in the tables, also Marvin Sketch. ”
The word “only” in the abstract has been removed. The spelling of temperature in the tables and Marvin Sketch have been corrected.